# Consumption of sorghum-based products and their impact on blood glucose and satiety in adult patients: A protocol for systematic review and meta-analysis

**Guilherme Augusto Loiola Passos**[ID]◉, **Érica Aguiar Moraes**◉, **Luís Carlos Lopes-Junior**[ID]◉*

Graduate Program in Nutrition and Health – PPGNS, Health Sciences Center at the Federal University of Espírito Santo (UFES), Vitória, ES, Brazil

◉ These authors contributed equally to this work.
* lopesjr.lc@gmail.com

## Abstract

### Background

The consumption of sorghum-based foods has been associated with reduced post-prandial blood glucose levels and increased satiety in previous studies. Sorghum's low glycemic index, high fiber content, and rich profile of bioactive compounds may contribute to improved health outcomes. Nutritional strategies incorporating sorghum could serve as a valuable tool for the prevention of diabetes, obesity, as well as other non-communicable diseases.

### Purpose

To identify and evaluate the evidence on the effectiveness of the sorghum-based foods in modulating blood glucose levels and promoting satiety in adults.

### Methods and expected outputs

This systematic review protocol was developed in accordance with the Preferred Reporting Items for Systematic Review and Meta-Analysis Protocols (PRISMA-P) guidelines. We will include randomized controlled trials (RCTs) and non-randomized controlled trials (NRCTs) published through February 2025 that assess the effects of sorghum-based food consumption. Qualitative studies, guidelines, and reviews will be excluded. Six electronic databases—MEDLINE/PubMed, Embase, Cochrane Central Register of Controlled Trials, Web of Science, Scopus, and ScienceDirect—will be systematically searched. Additional sources will include ClinicalTrials.gov, The British Library, Google Scholar, International Diabetes Federation, and American Diabetes Association (ADA). No restrictions on language or publication date will be applied. Two independent reviewers will perform study selection, data extraction, and

**Data availability statement:** All relevant data from this study will be made available upon study completion.

**Funding:** The author(s) received no specific funding for this work.

**Competing interests:** The authors have declared that no competing interests exist.

risk of bias assessment according to the study design. A qualitative synthesis will be presented. Statistical heterogeneity will be assessed using the I² statistic. If appropriate, a meta-analysis will be conducted using a random-effects model.

## Expected outcomes

To the best of our knowledge, this will be the first systematic review to specifically address the impact of sorghum-based food consumption on glycemic response and satiety. The findings are expected to provide robust evidence to inform future research and support nutritional strategies involving sorghum-based products for health promotion.

## PROSPERO registration ID

CRD42023431520

## 1 Introduction

Cereals and pseudocereals account for approximately 35% of daily energy intake worldwide [1]. Whole grain consumption has been consistently linked to health benefits, particularly in relation to type 2 diabetes mellitus (T2DM), weight management, and satiety [2]. These benefits are largely attributed to the higher content of dietary fiber, antioxidants, and phytochemicals found in whole grains [2]. Among cereals, sorghum (*Sorghum bicolor [L.] Moench)* has attracted increasing attention in recent years due to its nutritional profile and potential to support the development of healthy and functional food products [3]. Widely consumed in Africa and Asia, sorghum is a versatile cereal crop with favorable agronomic characteristics and a high concentration of bioactive compounds—including phenolics, anthocyanins, and condensed tannins—as well as fiber, resistant starch, proteins, lipids, carbohydrates, vitamins, and minerals [4,5].

The global rise in malnutrition has contributed to the growing epidemic of overweight and obesity worldwide. According to estimates, poor nutrition accounts for approximately 19% of all diseases and premature deaths globally [6,7]. Obesity, in particular, is a multifactorial condition primarily influenced by biological and environmental factors. A diet rich in energy-dense foods, combined with insufficient physical activity, promotes fat accumulation and increases body mass index (BMI), ultimately leading to obesity [8]. Furthermore, obesity is chronically associated with oxidative stress and reduced total antioxidant capacity, both of which significantly contribute to the development of diet-related non-communicable diseases, including diabetes mellitus (DM) [8–12].

The human appetite system is closely linked to body composition and, consequently, to the development of obesity. Appetite refers to the physiological need for energy intake (food consumption) and is often associated with motivational states such as hunger [13]. In this context, satiety plays a key role in obesity and type 2 diabetes mellitus (T2DM), as the sensation of "fullness" under

normal physiological conditions regulates appetite [14]. Importantly, hormonal mechanisms and the physical and chemical properties of foods influence satiety, affecting both short- and long-term satiety responses [15]. Therefore, enhancing the diet with foods that promote high satiation while maintaining a low glycemic index (GI) and glycemic load (GL) may represent an effective strategy for regulating energy intake and preventing weight gain, obesity, and T2DM [16].

Given the urgent need for effective nutritional strategies, sorghum stands out for its unique profile of phenolic compounds and condensed tannins found in certain sorghum genotypes, as well as its high fiber content. These characteristics provide notable health benefits, particularly by reducing oxidative stress, which plays a key role in the prevention of various non-communicable diseases [3]. Due to these favorable health properties and the richness of bioactive compounds within its food matrix, sorghum has been increasingly studied and incorporated into food products to assess its potential in improving health outcomes, including lowering blood glucose levels, enhancing satiety, supporting weight management, and preventing obesity [9].

Two previous systematic reviews, conducted by Ducksbury et al. [17] and Simnadis et al. [18], have examined the broader health effects of sorghum consumption. Simnadis et al. included observational and longitudinal studies to explore sorghum's role in chronic disease prevention and other health-related outcomes, while Ducksbury et al. focused on interventional studies assessing sorghum's effects on markers of chronic disease. Both reviews considered a broad and diverse population, without restrictions related to age, sex, health status, socioeconomic background, or geographic location.

In contrast, our systematic review is distinct in its specific focus on the effects of sorghum-based products on blood glucose regulation and satiety in adult patients. This targeted analysis addresses key components of metabolic health that are critical in the context of the global increase in non-communicable diseases.

This study employs systematic review methods to synthesize evidence, offering valuable insights into the effectiveness of sorghum-based dietary interventions for improving glycemic control and promoting satiety in adults. Ultimately, this review will contribute to advancing scientific knowledge by delivering a focused evaluation of the specific metabolic health benefits of sorghum consumption. The findings are expected to inform future dietary recommendations and interventions aimed at preventing chronic conditions such as type 2 diabetes mellitus (T2DM) and obesity.

## 2 Materials and methods

This systematic review protocol was developed in accordance with the Preferred Reporting Items for Systematic Reviews and Meta-Analyses Protocols-PRISMA-P [19]. Additionally, the protocol has been registered with the International Prospective Register of Systematic Reviews (PROSPERO) (registration ID: CRD42023431520).

### 2.1 Search strategy

The search strategy will be conducted across six electronic databases, covering all records from their inception through February 2025: I- Medical Literature Analysis and Retrieval System Online-MEDLINE/PubMed, II – Embase, III – The Cochrane Central Register of Controlled Trials, IV – Web of Science, V- Scopus, and VI – ScienceDirect. Additional sources will include trial registries such as Clinical trials.gov-NIH and WHO International Clinical Trials Registry Platform. Secondary searches will also be carried out in The British Library, Google Scholar, International Diabetes Federation (IDF) Congress Abstract Archive, American Diabetes Association (ADA) Scientific Sessions Abstract Database and medRXiv. No restriction will be applied regarding language or publication date. Furthermore, reference lists of all included articles will be manually screened to identify additional relevant studies [20]. The search strategy will be based on key terms derived from the PICOS (Population/Intervention/Comparison/Outcomes/Study Design) framework [21], which guided the development of our research question: What is the effectiveness of incorporating sorghum into the diet on (a) blood glucose levels and (b) satiety in adults? According to the PICOS structure,

the components are defined as follows: P = (Adult patients), I = (Sorghum-based food products), C = (Receiving other cereal non-sorghum), O = (Primary outcomes: blood glucose levels and satiety; Secondary outcomes: overweight and obesity), S = RCT or NRCT).

Initially, controlled vocabulary (e.g., MeSH/DeCS terms) and their respective synonyms will be identified, and relevant keywords will be selected. Boolean operators "AND" and "OR" will be used to combine search terms and refine the strategy [22,23]. The preliminary search strategy for MEDLINE/PubMed, incorporating MeSH terms, synonyms, and keywords, is presented in Table 1.

## 2.2 Eligibility criteria

All randomized controlled trials (RCTs) and non-randomized controlled trials (NRCTs) published up to the end of February 2025 that examine the acute or chronic effects of consuming sorghum-based foods will be included.

## 2.3 Population

- Inclusion criteria: healthy young adults and adults of both sexes, age > 18 years, of any ethnicity, diagnosed with chronic disease.

**Table 1. Preliminary pilot search strategy in MEDLINE/PubMed.**

| Database | Search strategy |
|---|---|
| MEDLINE/Pubmed | **#1:** ("Adult"[Mesh terms] OR "Adults"[All fields] OR "Young Adult"[Mesh terms] OR "Adult, Young"[All fields] OR "Adults, Young"[All fields] OR "Young Adults"[All fields] OR "Middle Aged"[Mesh terms] OR "Middle Age"[All fields] OR "Aged"[Mesh terms] OR "Elderly"[All fields]OR "Aged, 80 and over"[Mesh terms] OR "Oldest Old"[All fields] OR "Research Subjects"[Mesh terms] OR "Research Subject"[All Fields] OR"Subject, Research" [All Fields] OR "Subjects, Research" [All Fields] OR "Human Subjects"[All Fields] OR "Human Subject"[All Fields] OR "Subject, Human" [All Fields] OR "Subjects, Human" [All Fields]) |
| | **#2:** ("Sorghum"[Mesh terms] OR "Sorghums"[All fields] OR "Sorghum bicolor"[All fields] OR "bicolor, Sorghum"[All fields]) |
| | **#3:** #1 AND #2 |
| | **#4:** ("Non Sorghum"[All fields] OR "Avena"[Mesh terms] OR "Avenas"[All fields] OR "Oats"[All fields] OR "Oat"[All fields] OR "Avena sativa"[All fields] OR "Millets"[Mesh terms] OR "Millet"[All fields] OR "Oryza"[Mesh terms] OR "Rice"[All fields] OR "Rices"[All fields] OR "Oryza sativa"[All fields] OR "Secale"[Mesh terms] OR "Secale cereale"[All fields] OR "Rye"[All fields] OR "Ryes"[All fields] OR "Triticale"[Mesh terms] OR "Triticosecale"[All fields] OR "Triticum x Secale"[All fields] OR "Triticum"[Mesh terms] OR "Triticum turgidum"[All fields] OR "Wheat"[All fields] OR "Triticum aestivum"[All fields] OR "Triticum vulgare"[All fields] OR "Triticum spelta"[All fields] OR "Durum Wheat"[All fields] OR "Durum Wheats"[All fields] OR "Wheat, Durum"[All fields] OR "Triticum turgidum subsp. durum"[All fields] OR "Triticum durum"[All fields] OR "Zea mays"[Mesh terms] OR "Zea"[All fields] OR "Corn"[All fields] OR "Indian Corn"[All fields] OR "Maize"[All fields] OR "Teosinte"[All fields]) |
| | **#5:** #1 AND #2 |
| | **#6:** ("Blood Glucose"[Mesh terms] OR "Blood Glucose"[All fields] OR "Blood Sugar"[All fields] OR "Sugar, Blood"[All fields] OR "Glucose, Blood"[All fields] OR "Glycemic Control"[Mesh terms] OR "Glycemic Control"[All fields] OR "Glycaemic Control"[All fields] OR "Control, Glycemic"[All fields] OR "Blood Glucose Control"[All fields] OR "Control, Blood Glucose"[All fields] OR "Glucose Control, Blood"[All fields] OR "Satiation"[Mesh terms] OR "Satiation"[All fields] OR "Satiety Response"[Mesh terms] OR "Satiety Response"[All fields] OR "Satiations"[All fields] OR "Response, Satiety"[All fields] OR "Responses, Satiety"[All fields] OR "Satiety Responses"[All fields] OR "Obesity"[Mesh terms] OR "Body Weight"[Mesh terms] OR "Body Weights"[All fields] OR "Weight, Body"[All fields] OR "Weights, Body"[All fields] OR "Overweight"[Mesh terms] OR "Weight Gain"[Mesh terms] OR "Gain, Weight"[All fields] OR "Gains, Weight"[All fields] OR "Weight Gains"[All fields] OR "Weight Loss"[Mesh terms] OR "Loss, Weight" OR "Losses, Weight"[All fields] OR "Weight Losses"[All fields] OR "Weight Reduction"[All fields] OR "Reduction, Weight"[All fields] OR "Reductions, Weight"[All fields] OR "Weight Reductions"[All fields]) |
| | **#7:** #5 AND #6 |
| | **#8:** (randomized controlled trial[pt] OR controlled clinical trial[pt] OR randomized[All fields] OR placebo[All fields] OR clinical trials as topic[mesh:noexp] OR randomly[All fields] OR trial[ti] NOT (animals[mh] NOT humans [mh])) |
| | **#9:** #7 AND #8 |

• Exclusion criteria: children and adolescents, and adults with acute infectious conditions.

### 2.4 Intervention/exposure

• Inclusion criteria: Adults consuming all types of sorghum-based products, including both processed and whole grains forms as part of the intervention.

  Exclusion criteria: Studies in which participants consume non-sorghum food products as part of the intervention.

### 2.5 Outcomes

• Inclusion criteria: Studies reporting the effects of sorghum-based food consumption, preferably assessed using the following validated parameters:

  • **Blood Glucose Regulation:** Fasting blood glucose, glycated hemoglobin (HbA1c), Homeostatic Model Assessment of Insulin Resistance (HOMA-IR).

  • **Satiety:** Visual Analogue Scale (VAS), Ad Libitum Intake (ALI), glucagon-like peptide-1 (GLP-1), leptin, peptide YY (PYY), ghrelin.

  • **Overweight and Obesity:** Body Mass Index (BMI).

  While these are the preferred assessment tools, other validated measures may be considered if justified and relevant to the research question.

• Exclusion criteria: Studies assessing the effects of non-sorghum-based products or sorghum combined with other cereals.

### 2.6 Studies

• Inclusion criteria: RCT and NRCT.

• Exclusion criteria: Qualitative studies, animal studies, studies involving individuals under 18 years of age, guidelines, conference abstracts, and review articles.

  Additionally, reference lists of all included studies will be manually screened to identify any further relevant publications. No restrictions will be applied regarding language or publication date in the search strategy. The review team is fluent in English, Portuguese, and Spanish. For articles published in other languages, translation support will be obtained from the Faculty of Letters and the Graduate Program in Linguistics at the affiliated university, who will assist with the translation of eligible studies to ensure their inclusion in the review.

### 2.7 Study selection

Initially, all records retrieved from the six electronic databases will be imported into EndNote™, in accordance with the PICOS strategy. Duplicate entries will be identified and removed. Two independent reviewers will then screen the titles and abstracts using the Rayyan™ application. Following this initial screening, the full texts of potentially eligible studies will be retrieved and assessed independently by the same two reviewers based on the predefined inclusion and exclusion criteria, with the goal of minimizing selection bias. Any discrepancies between reviewers will be resolved through discussion and, if necessary, consultation with a third reviewer to reach consensus. The study selection process will be documented and illustrated using a flow diagram in accordance with the PRISMA 2020 statement [24], will be provided (Fig 1).

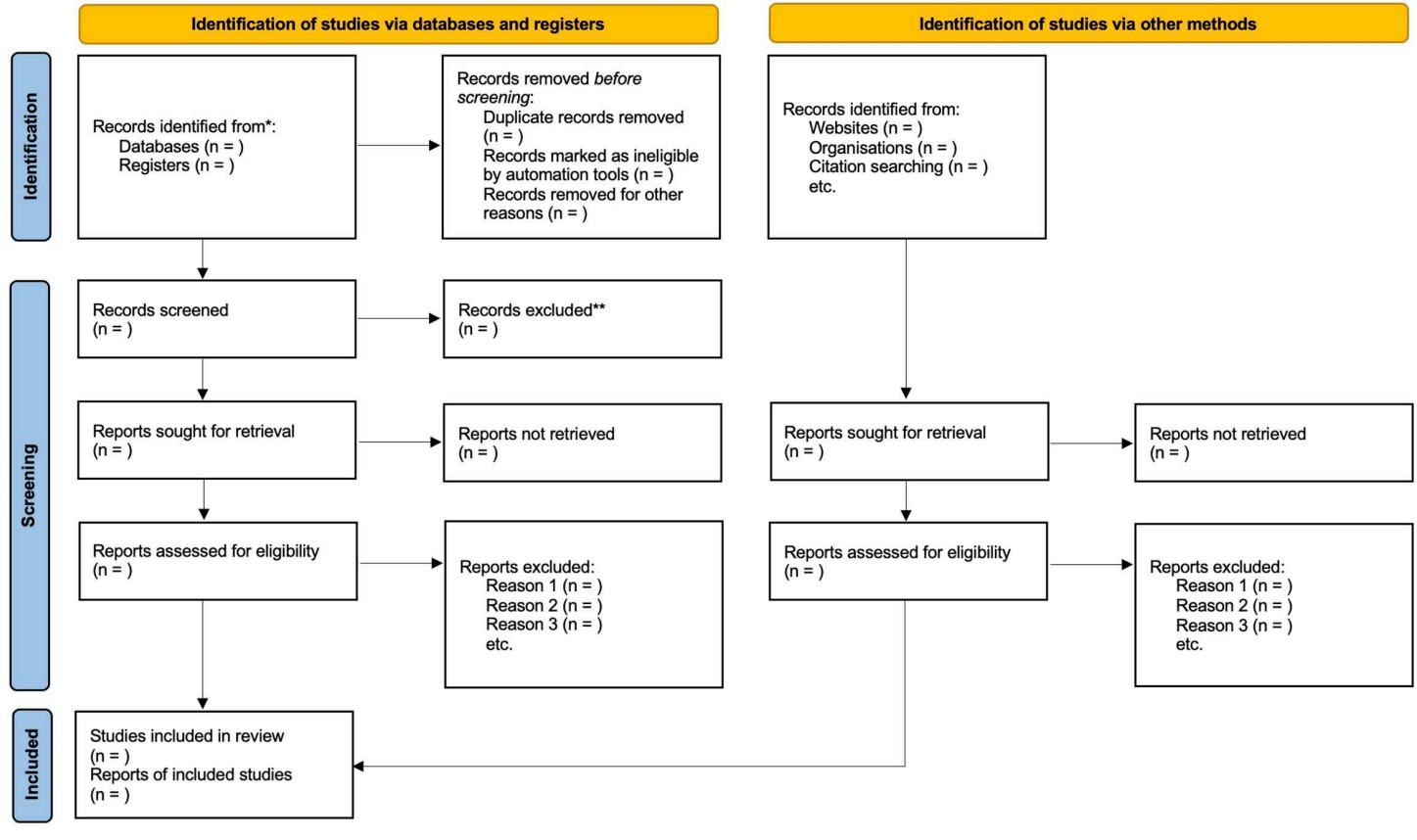

**Fig 1. PRISMA flowchart.**

## 2.8 Data extraction and data synthesis

Two reviewers will independently extract data from each included study using standardized forms previously published in the literature [20,23,25–27]. The expected completion date for this systematic review is September 30, 2025. In addition, upon completion of the review, all extracted data and analysis code will be made publicly available in the Open Science Framework (OSF) repository. The direct link to the repository will be provided in the final publication.

The information to be extracted is showed in Chart 1:

## 2.9 Critical appraisal of the studies included

First, the level of evidence for each included study will be determined based on study design, following the established evidence hierarchy: levels I and II will be considered strong, levels III to V as moderate, and levels VI to VII as weak [28]. Second, the internal validity and risk of bias of randomized controlled trials (RCTs) will be assessed using the revised Cochrane Risk of Bias tool for randomized trials (RoB 2) [29]. For non-randomized controlled trials (NRCTs), the Risk of Bias in Non-randomized Studies of Interventions (ROBINS-I) tool will be applied [30]. The same two reviewers will independently conduct the critical appraisal of all included studies.

## 2.10 Data synthesis and statistical analysis

The characteristics of the included studies will be summarized and presented in tables. Results will be organized according to study design, and a qualitative synthesis of the data will be provided. To assess heterogeneity among studies, the I²

| Study: | Critical appraisal: |
|---|---|
| **Study characteristics** | |
| Authors | |
| Title | |
| Year of publication and doi | |
| Country(ies) where study is based | |
| Conflicts of interests | |
| Sponsorship | |
| Background | |
| Rationale | |
| Hypothesis | |
| Objectives | |
| **Methods** | |
| Methodology is reported is in compliance with CONSORT (clinical trials)<br>(  ) Yes<br>(  ) No<br>(  ) Partially | |
| Study design<br><br>For experimental or quase-experimental study | **a)** Trial Register:<br>**b)** Trial arms:<br>- Experimental Group:<br>- Control Group<br>**c)** Randomization:<br>**d)** Masking:<br>**e)** Intervention protocol:<br>**f)** Per-protocol and modified intention-to-treat analyses:<br>  - Per-protocol:<br>  - Intention-to-treat:<br>  - Dropouts: |
| Setting | |
| Sample size: | |
| Inclusion criteria (definition of intervention interest) | |
| Exclusion criteria | |
| Confounding factors/Interaction factors considered | |
| Ethical aspects | |
| Procedure for data collection: | |
| Instruments for data collection | |
| Outcomes | - Primary outcomes: Impact on blood glucose levels and satiety;<br>- Secondary outcomes: overweight and obesity |
| Follow-up | |
| Statistical analysis | |
| **Results** | |
| Main results | |
| Clinical significance | |
| Limitations of the study | |
| Strengths of the study | |
| **Conclusions** | |
| Main conclusions | |
| Implication for clinical practice | |

**Chart 1. The data extraction form.**

statistic will be used to estimate the percentage of total variation attributable to heterogeneity rather than chance [31,32]. Based on the degree of heterogeneity ($I^2$), we will determine whether conducting a meta-analysis is appropriate [33]. If feasible, a meta-analysis will be performed using a random-effects model.

Subgroup analyses will also be conducted using random-effects models, adjusted for age, sex, type of food product, quantity of sorghum consumed, and the presence of overweight or obesity. Pooled effect estimates will be calculated with 95% confidence intervals, and statistical significance will be set at an alpha level of 0.05. All analyses will be conducted using SPSS version 28.0. Given the exploratory nature of this review and the limited number of pooled comparisons expected, we do not plan to apply formal corrections for multiple testing. All secondary analyses will be interpreted with caution and considered hypothesis-generating

The certainty of the evidence will be assessed according to Cochrane recommendations and rated using the Grading of Recommendations, Assessment, Development, and Evaluation (GRADE) approach [34]. To evaluate potential publication bias, funnel plots will be visually inspected. Additionally, the trim-and-fill method proposed by Duval and Tweedie [35] will be applied to estimate the number of potentially missing studies and assess the adjusted effect size after imputation. The presence of asymmetry will also be statistically tested using Egger's test [36].

### 2.11 Ethics issues and dissemination

No ethical approval is required for this study design, as it is based solely on the analysis of previously published data. The systematic review will be reported in accordance with the PRISMA 2020 statement [24]. The findings of this review will be disseminated through publication in peer-reviewed journals and presentations at national and international scientific conferences.

## 3 Discussion

A clearer understanding of how sorghum-based foods affect blood glucose response and satiety may help identify which specific components of this cereal exert the greatest influence on these outcomes. Such insights can inform the development of targeted nutritional strategies using sorghum-based foods to support the prevention of chronic diseases, including type 2 diabetes mellitus (T2DM) and obesity. Furthermore, the results of this systematic review may contribute to future research by offering more precise evidence regarding the effectiveness of sorghum in the context of metabolic health.

This future systematic review presents several strengths that enhance its relevance and methodological rigor. It will be the first to focus specifically on the effects of sorghum-based food consumption on blood glucose regulation and satiety in adults, addressing a critical gap in the current literature. The protocol adheres to established methodological standards, including the PRISMA-P checklist, and employs validated tools such as RoB 2 and ROBINS-I for risk of bias assessment. The comprehensive and unrestricted search strategy, as well as the inclusion of both randomized and non-randomized trials, further strengthens the review's breadth and potential impact. However, several limitations are anticipated. These include the potential scarcity of high-quality studies with strong methodological designs, small sample sizes, and heterogeneity among studies in terms of populations, interventions, and outcome measures. Such factors may compromise the robustness of the findings and reduce the overall certainty of the evidence. Despite these challenges, the systematic review will aim to address these issues through rigorous data synthesis, transparent reporting, and the application of the GRADE approach to evaluate the certainty of the evidence, ultimately providing a balanced and critical appraisal of the available research.

## Supporting information

**S1 File. PRISMA-P 2015 checklist.**
(DOCX)

## Author contributions

**Conceptualization:** Guilherme Augusto Loiola Passos, Érica Aguiar Moraes, Luís Carlos Lopes-Júnior.

**Data curation:** Guilherme Augusto Loiola Passos, Érica Aguiar Moraes, Luís Carlos Lopes-Júnior.

**Formal analysis:** Guilherme Augusto Loiola Passos, Érica Aguiar Moraes, Luís Carlos Lopes-Júnior.

**Funding acquisition:** Luís Carlos Lopes-Júnior.

**Investigation:** Érica Aguiar Moraes, Luís Carlos Lopes-Júnior.

**Methodology:** Luís Carlos Lopes-Júnior.

**Project administration:** Luís Carlos Lopes-Júnior.

**Resources:** Luís Carlos Lopes-Júnior.

**Software:** Guilherme Augusto Loiola Passos, Luís Carlos Lopes-Júnior.

**Supervision:** Luís Carlos Lopes-Júnior.

**Validation:** Érica Aguiar Moraes, Luís Carlos Lopes-Júnior.

**Visualization:** Guilherme Augusto Loiola Passos, Érica Aguiar Moraes, Luís Carlos Lopes-Júnior.

**Writing – original draft:** Guilherme Augusto Loiola Passos, Érica Aguiar Moraes, Luís Carlos Lopes-Júnior.

**Writing – review & editing:** Guilherme Augusto Loiola Passos, Luís Carlos Lopes-Júnior.

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
