## [Decision Letter · Decision Letter 0]

5 May 2025

PONE-D-25-15492Consumption of sorghum-based products and their impact on blood glucose and satiety in adult patients: a protocol for systematic review and meta-analysisPLOS ONE

Dear Dr. Lopes-Júnior,

Thank you for submitting your manuscript to PLOS ONE. After careful consideration, we feel that it has merit but does not fully meet PLOS ONE’s publication criteria as it currently stands. Therefore, we invite you to submit a revised version of the manuscript that addresses the points raised during the review process.

We look forward to receiving your revised manuscript.

Kind regards,

Shailender Kumar Verma, Ph.D.

Academic Editor

PLOS ONE

Journal Requirements:

Reviewers' comments:

Reviewer's Responses to Questions

**Comments to the Author**

1. Does the manuscript provide a valid rationale for the proposed study, with clearly identified and justified research questions?

Reviewer #1: Yes

Reviewer #2: Yes

Reviewer #3: Yes

Reviewer #4: Yes

2. Is the protocol technically sound and planned in a manner that will lead to a meaningful outcome and allow testing the stated hypotheses?

Reviewer #1: Yes

Reviewer #2: Yes

Reviewer #3: Yes

Reviewer #4: Yes

3. Is the methodology feasible and described in sufficient detail to allow the work to be replicable?

Reviewer #1: Yes

Reviewer #2: Yes

Reviewer #3: Yes

Reviewer #4: Yes

4. Have the authors described where all data underlying the findings will be made available when the study is complete?

Reviewer #1: Yes

Reviewer #2: Yes

Reviewer #3: No

Reviewer #4: Yes

5. Is the manuscript presented in an intelligible fashion and written in standard English?

Reviewer #1: Yes

Reviewer #2: Yes

Reviewer #3: Yes

Reviewer #4: Yes

6. Review Comments to the Author

You may also provide optional suggestions and comments to authors that they might find helpful in planning their study.

Reviewer #1: Interesting study. Sound methodology. I agree with the authors that a lack of published material may limit the outcome, but at least this provides a sound basis for further empiric study.

Minor points:

1. Reframe your research question as 2 separate ones. Blood glucose and satiety will probably be linked, but not necessarily.

In addition, why include the word 'patients' in your research question? Shouldn't you include well individuals? If you just want participants who are ill, then specify the illness (e.g. Type 2 DM).

I would suggest revision to the following: "What is the effectiveness of incorporating sorghum into the diet on (a) blood glucose levels and (b) satiety in adults?

2. P5, line 158, the sentence "In line with PICOS strategy, our research question is: " is not completed (but is answered before this in the paragraph)

Reviewer #2: This a systemic review protocol paper outlining the strategy for a future systemic review project on the role of sorghum on satiety and blood glucose. I note that this is a a revision and I was not one of the original reviewers. My own reading of the paper is that it is acceptable for publication based on its ability to meet the methodological requirements of a planned protocol for a systemic review. The Introduction gives a satisfactory background justifying the study. With repect to the previous reviews, the aiuthors have made the requisite corrections and have replied to the complaint of the last reviewer.

Reviewer #3: Suggestions for Improvement

1. Eligibility Criteria: Please provide the thought processes in the creation of the inclusion and exclusion criteria. Specifically, clarify the rationale for excluding older adults (>65 years) with diabetes/chronic diseases. Provide literature support for this decision, as this population may benefit from sorghum-based interventions.

2. Data Availability and Reproducibility: While the protocol states data will be made available upon completion, specifying a repository (e.g., GitHub, OSF) for extracted data and analysis code would align with best practices for open science. This is particularly relevant since the data are derived from publicly available studies.

3. Active Voice Revision: The manuscript occasionally uses passive voice (e.g., Line 128, p. 4). Revising to active voice (e.g., *"This study employs systematic review methods to synthesize evidence..."*) would improve clarity, as recommended by the [CDC’s writing guidelines](https://www.cdc.gov/pcd/issues/2018/18_0085.htm).

4. Pilot Testing of Search Strategy: Consider running a trial search to ensure inclusion criteria are not overly restrictive, especially if the authors aim to capture key studies (e.g., prior reviews or known manuscripts).

5. Optional Recommendations:

1. Publication of Analysis Code: Sharing statistical code (e.g., R/Python scripts for meta-analysis) as a supplement or in a repository would enhance reproducibility.

Overall Recommendation

This protocol is well-conceived and merits publication after addressing the minor revisions above. The study has the potential to inform dietary guidelines and future research on sorghum’s health benefits. I commend the authors for their thorough approach and look forward to the completed review.

Reviewer #4: In general, this seems to be a reasonable protocol for a systematic review of the topic chosen by the authors. My main concern is the lack of publications on this topic present in PubMed. If the majority of the analysis will be reliant on reports in clinical trials databases and secondary sources that are not peer-reviewed, this will result in a review with weak conclusions. It would make sense to at least identify additional appropriate conference abstract databases that could be included in the analysis, e.g. International Diabetes Federation, American Diabetes Association, etc.

Specific points about manuscript:

• Search term #1 appears to be at odds with the exclusion criteria – why are you searching for “aged” subjects when these are excluded from the review?

• Search term #6 should include the variant spelling “glycaemic” as this spelling was widely used until recently.

• There is no evidence anywhere that the authors will correct for multiple testing within their statistical analysis. If they are deliberately not going to do that, then some justification for omission of this step must be provided.

7. PLOS authors have the option to publish the peer review history of their article (what does this mean? ). If published, this will include your full peer review and any attached files.

**Do you want your identity to be public for this peer review?** For information about this choice, including consent withdrawal, please see our Privacy Policy .

Reviewer #1: **Yes: ** Professor Neil Kennedy

Reviewer #2: No

Reviewer #3: **Yes: ** Anthony Onde Morada, MD

Reviewer #4: No

---

## [Author Response · Author response to Decision Letter 1]

29 May 2025

Response to Reviewers

PLOS ONE Decision: Revision required [PONE-D-25-15492]

Vitória, ES, Brazil, May 28th, 2025

Dear Federico Vita,

Academic Editor.

We sincerely thank the editors and reviewers for their valuable comments and suggestions regarding the structure and content of our manuscript. We have carefully addressed each point raised and provide a detailed, item-by-item description of the changes made below.

We are grateful for the thoughtful and constructive feedback, which has greatly enhanced the quality and clarity of our work. We fully agree with all the comments and have incorporated the recommended revisions throughout the manuscript.

Below, we provide detailed responses to each of the editor’s comments.

Reviewer #1: Major Revision

Interesting study. Sound methodology. I agree with the authors that a lack of published material may limit the outcome, but at least this provides a sound basis for further empiric study.

Response: We sincerely appreciate your positive feedback and thoughtful evaluation of our manuscript.

Minor points:

1. Reframe your research question as 2 separate ones. Blood glucose and satiety will probably be linked, but not necessarily.

In addition, why include the word 'patients' in your research question? Shouldn't you include well individuals? If you just want participants who are ill, then specify the illness (e.g. Type 2 DM).

I would suggest revision to the following: "What is the effectiveness of incorporating sorghum into the diet on (a) blood glucose levels and (b) satiety in adults?”

Response: We thank the reviewer for the thoughtful and constructive feedback. In response to the suggestion, we have revised the research question to clearly separate the outcomes and to use terminology that more accurately reflects the study population. The updated version has been incorporated into the manuscript accordingly.

2. P5, line 158, the sentence "In line with PICOS strategy, our research question is: " is not completed (but is answered before this in the paragraph)

Response: We appreciate your careful reading of our manuscript. The excerpt in question has been removed from the text.

Reviewer #2:

This is a systematic review protocol paper outlining the strategy for a future systematic review project on the role of sorghum on satiety and blood glucose. I note that this is a revision and I was not one of the original reviewers. My own reading of the paper is that it is acceptable for publication based on its ability to meet the methodological requirements of a planned protocol for a systematic review. The Introduction gives a satisfactory background justifying the study. With respect to the previous reviews, the authors have made the requisite corrections and have replied to the complaint of the last reviewer.

Response: We sincerely appreciate your thoughtful evaluation and positive feedback on our manuscript.

Reviewer #3: Suggestions for Improvement

1. Eligibility Criteria: Please provide the thought processes in the creation of the inclusion and exclusion criteria. Specifically, clarify the rationale for excluding older adults (>65 years) with diabetes/chronic diseases. Provide literature support for this decision, as this population may benefit from sorghum-based interventions.

Response: We are truly grateful for your thoughtful evaluation of our manuscript. This was indeed an oversight, and we sincerely appreciate you bringing it to our attention. Older adults are not excluded from our review and are, in fact, included in the search strategy, as this population - particularly those with chronic conditions - may benefit from sorghum-based dietary interventions. The excerpt suggesting their exclusion was removed to reflect our eligibility criteria accurately.

Now reads, line 173: “Exclusion criteria: children, adolescents, and adults (age > 18 years of any ethnicity, diagnosed with diabetes mellitus or any other chronic disease).”

2. Data Availability and Reproducibility: While the protocol states data will be made available upon completion, specifying a repository (e.g., GitHub, OSF) for extracted data and analysis code would align with best practices for open science. This is particularly relevant since the data are derived from publicly available studies.

Response: We sincerely thank the reviewer for raising this important point regarding data sharing and reproducibility. In response, we have now specified the repository where the data and analysis code will be made available upon completion of the study. Specifically, we will use the Open Science Framework (OSF), a platform widely adopted in open science practices.

Accordingly, we have revised the “Data Availability” statement in the manuscript to read as follows:

“Upon completion of the review, all extracted data and analysis code will be made publicly available in the Open Science Framework (OSF) repository. The direct link to the repository will be provided in the final publication.”

We trust this modification addresses the reviewer’s concerns and strengthens the transparency and reproducibility of our study.

3. Active Voice Revision: The manuscript occasionally uses passive voice (e.g., Line 128, p. 4). Revising to active voice (e.g., *"This study employs systematic review methods to synthesize evidence..."*) would improve clarity, as recommended by the [CDC’s writing guidelines] (https://www.cdc.gov/pcd/issues/2018/18_0085.htm).

Response: The suggestion has been accepted and incorporated into the revised manuscript.

Now reads, line 129: “This study employs systematic review methods to synthesize evidence, offering valuable insights into the effectiveness of sorghum-based dietary interventions for improving glycemic control and promoting satiety in adults.”

4. Pilot Testing of Search Strategy: Consider running a trial search to ensure inclusion criteria are not overly restrictive, especially if the authors aim to capture key studies (e.g., prior reviews or known manuscripts).

Response: We appreciate the suggestion. A pilot testing of the search strategy was conducted during the initial stages of the protocol to ensure the inclusion criteria were appropriate and not overly restrictive. This preliminary test helped confirm the strategy’s ability to retrieve key studies and manuscripts in the field.

5. Optional Recommendations:

1. Publication of Analysis Code: Sharing statistical code (e.g., R/Python scripts for meta-analysis) as a supplement or in a repository would enhance reproducibility.

Response: We fully agree with the reviewer that sharing the statistical code is essential for ensuring reproducibility and transparency. As noted, all analysis scripts used for data synthesis (e.g., SPSS syntax files, R/Python scripts, or any additional computational tools applied) will be uploaded to the Open Science Framework (OSF) repository alongside the dataset upon completion of the review.

To reflect this commitment, we have further revised the "Data Availability" statement to explicitly mention the inclusion of statistical code. The revised version now reads:

“Upon completion of the review, all extracted data and statistical analysis code (e.g., SPSS, R, or Python scripts) will be made publicly available in the Open Science Framework (OSF) repository. The direct link to the repository will be provided in the final publication.”

We hope this revision adequately addresses the reviewer’s valuable suggestion and demonstrates our commitment to open science practices.

Overall Recommendation

This protocol is well-conceived and merits publication after addressing the minor revisions above. The study has the potential to inform dietary guidelines and future research on sorghum’s health benefits. I commend the authors for their thorough approach and look forward to the completed review.

Response: We are truly grateful for your thoughtful evaluation and constructive feedback, which have greatly contributed to the improvement of our manuscript.

Reviewer #4: Major comments

In general, this seems to be a reasonable protocol for a systematic review of the topic chosen by the authors. My main concern is the lack of publications on this topic present in PubMed. If the majority of the analysis will be reliant on reports in clinical trials databases and secondary sources that are not peer-reviewed, this will result in a review with weak conclusions. It would make sense to at least identify additional appropriate conference abstract databases that could be included in the analysis, e.g. International Diabetes Federation, American Diabetes Association, etc.

Response:

We thank the reviewer for this thoughtful and constructive comment. We acknowledge that the current body of peer-reviewed literature on this specific topic may be limited. To mitigate this, and in line with the reviewer’s excellent suggestion, we have revised our protocol to include searches in the abstract repositories of major scientific organizations relevant to the scope of our study.

In particular, we have now added the following sources to our search strategy:

• International Diabetes Federation (IDF) Congress Abstract Archive

• American Diabetes Association (ADA) Scientific Sessions Abstract Database

These databases will be searched systematically to identify any additional unpublished or recent studies, abstracts, or relevant data that may not yet be indexed in PubMed or other traditional databases. This amendment aims to improve the comprehensiveness and inclusiveness of our evidence base and reduce the risk of publication bias.

We have also updated the "Search Strategy" section of the manuscript to reflect these additions.

Specific points about manuscript:

Search term #1 appears to be at odds with the exclusion criteria – why are you searching for “aged” subjects when these are excluded from the review?

Response:

We thank the reviewer for this pertinent and helpful observation. To clarify, older adults are not excluded from our review. The inclusion of the term “aged” in the search strategy is therefore consistent with our eligibility criteria, which encompass adults aged 18 years and older, regardless of upper age limit.

We have reviewed the manuscript and revised any wording that may have implied the exclusion of older adults in order to ensure internal consistency throughout the text.

We are grateful to the reviewer for highlighting this important point, which has contributed to improving the clarity and methodological transparency of our protocol.

Search term #6 should include the variant spelling “glycaemic” as this spelling was widely used until recently.

Response: The suggestion has been accepted, and the text has been incorpored.

- There is no evidence anywhere that the authors will correct for multiple testing within their statistical analysis. If they are deliberately not going to do that, then some justification for omission of this step must be provided.

Response: We thank the reviewer for this insightful and important observation. At this stage, we anticipate conducting a limited number of primary and secondary analyses, guided by our pre-specified outcomes (i.e., blood glucose regulation and satiety, with overweight and obesity as secondary outcomes). Given the exploratory nature of this systematic review and the expectation of a small number of pooled comparisons, we do not plan to implement formal corrections for multiple testing (e.g., Bonferroni).

However, we recognize the risk of Type I error inflation and will interpret all secondary analyses with appropriate caution, clearly distinguishing between confirmatory and exploratory findings. This decision is in line with recommendations from the Cochrane Handbook and recent literature that advise against automatic adjustment in the context of systematic reviews with limited and clinically heterogeneous data.

To clarify this point, we have added the following sentence to the "Statistical Analysis" subsection of the manuscript:

“Given the exploratory nature of this review and the limited number of pooled comparisons expected, we do not plan to apply formal corrections for multiple testing. All secondary analyses will be interpreted with caution and considered hypothesis-generating.”

We are grateful to the reviewer for prompting this clarification, which strengthens the transparency of our statistical approach.

Best regards,

The authors.

---

## [Decision Letter · Decision Letter 1]

8 Jun 2025

PONE-D-25-15492R1Consumption of sorghum-based products and their impact on blood glucose and satiety in adult patients: a protocol for systematic review and meta-analysisPLOS ONE

Dear Dr. Lopes-Júnior,

Thank you for submitting your manuscript to PLOS ONE. After careful consideration, we feel that it has merit but does not fully meet PLOS ONE’s publication criteria as it currently stands. Therefore, we invite you to submit a revised version of the manuscript that addresses the points raised during the review process.

We look forward to receiving your revised manuscript.

Kind regards,

Shailender Kumar Verma, Ph.D.

Academic Editor

PLOS ONE

Reviewers' comments:

Reviewer's Responses to Questions

**Comments to the Author**

1. Does the manuscript provide a valid rationale for the proposed study, with clearly identified and justified research questions?

Reviewer #1: Yes

Reviewer #3: Yes

2. Is the protocol technically sound and planned in a manner that will lead to a meaningful outcome and allow testing the stated hypotheses?

Reviewer #1: Yes

Reviewer #3: Partly

3. Is the methodology feasible and described in sufficient detail to allow the work to be replicable?

Reviewer #1: Yes

Reviewer #3: Yes

4. Have the authors described where all data underlying the findings will be made available when the study is complete?

Reviewer #1: Yes

Reviewer #3: Yes

5. Is the manuscript presented in an intelligible fashion and written in standard English?

Reviewer #1: Yes

Reviewer #3: Yes

6. Review Comments to the Author

You may also provide optional suggestions and comments to authors that they might find helpful in planning their study.

Reviewer #1: Thanks for your positive response to the comments. All of my points have been adequately addressed. Good luck with your study.

Reviewer #3: 1. Eligibility criteria: The authors have partially addressed my concerns regarding population inclusion and exclusion criteria. However, there is a inconsistency that must be resolved:

a. Major error in exclusion criteria: The revised manuscript (line 178) states that "adults (age > 18 years, of any ethnicity, diagnosed with diabetes mellitus or any other chronic disease)" will be excluded. This creates a fundamental contradiction with multiple elements of the study:

- The study rationale emphasizes sorghum's benefits for metabolic conditions

- The search strategy (Table 1) includes all adult patients without any exclusion

- The authors' response to reviewers states that adults with chronic conditions are included

b. Required resolution: The authors must either:

- Correct what appears to be a placement error and clarify that adults with chronic conditions will be included (consistent with their stated rationale and search strategy), OR

- If exclusion is intended, provide clear justification and revise the title, abstract, search strategy, and study rationale to reflect a healthy-adults-only population.

2. Data Availability and Reproducibility: The authors adequately addressed my concerns by specifying use of the OSF repository, which is appropriately reflected in the manuscript.

3. Active Voice: While the authors incorporated active voice in line 133, this suggestion applies more broadly to improve manuscript readability. Multiple sections would benefit from active voice construction, particularly when describing researcher actions. Although passive voice serves appropriate functions (e.g., general statements or when the actor is unknown/insignificant), active voice typically enhances clarity in methodological descriptions. While not essential for approval, consistent voice improvement would enhance reader comprehension.

7. PLOS authors have the option to publish the peer review history of their article (what does this mean? ). If published, this will include your full peer review and any attached files.

**Do you want your identity to be public for this peer review?** For information about this choice, including consent withdrawal, please see our Privacy Policy .

Reviewer #1: **Yes: ** Prof Neil Kennedy, Queen's University Belfast

Reviewer #3: **Yes: ** Anthony Onde Morada

---

## [Author Response · Author response to Decision Letter 2]

12 Jun 2025

Response to Reviewers

PLOS ONE Decision: Revision required [PONE-D-25-15492]

Vitória, ES, Brazil, June 12th, 2025

Dear Federico Vita,

Academic Editor.

We sincerely thank the editors and reviewers for their valuable comments and suggestions regarding the structure and content of our manuscript. We have carefully addressed each point raised and provide a detailed, item-by-item description of the changes made below.

We are grateful for the thoughtful and constructive feedback, which has greatly enhanced the quality and clarity of our work. We fully agree with all the comments and have incorporated the recommended revisions throughout the manuscript.

Below, we provide detailed responses to each of the editor’s comments.

Reviewer #1: Thanks for your positive response to the comments. All of my points have been adequately addressed. Good luck with your study.

Response: We sincerely appreciate your positive feedback and thoughtful evaluation of our manuscript.

Reviewer #3: 1. Eligibility criteria: The authors have partially addressed my concerns regarding population inclusion and exclusion criteria. However, there is a inconsistency that must be resolved:

a. Major error in exclusion criteria: The revised manuscript (line 178) states that "adults (age > 18 years, of any ethnicity, diagnosed with diabetes mellitus or any other chronic disease)" will be excluded. This creates a fundamental contradiction with multiple elements of the study:

- The study rationale emphasizes sorghum's benefits for metabolic conditions

- The search strategy (Table 1) includes all adult patients without any exclusion

- The authors' response to reviewers states that adults with chronic conditions are included

b. Required resolution: The authors must either:

- Correct what appears to be a placement error and clarify that adults with chronic conditions will be included (consistent with their stated rationale and search strategy), OR

- If exclusion is intended, provide clear justification and revise the title, abstract, search strategy, and study rationale to reflect a healthy-adults-only population.

Response: We thank the reviewer for their careful reading and helpful comments.

We acknowledge the inconsistency noted in line 178 and confirm that this was indeed a typographical and placement error. Adults with chronic conditions such as diabetes mellitus are intended to be included in the study, as originally described in our rationale, search strategy (Table 1), and previous responses.

We have corrected the manuscript to reflect the accurate eligibility criteria, ensuring consistency across the rationale, methods, and inclusion/exclusion sections. The revised sentence now reads:

“Exclusion criteria will children and adolescents, and adults with acute infectious conditions.”

We appreciate the reviewer’s attention to this detail and the opportunity to clarify this important point.

2. Data Availability and Reproducibility: The authors adequately addressed my concerns by specifying use of the OSF repository, which is appropriately reflected in the manuscript.

Response: OK. Thank you.

3. Active Voice: While the authors incorporated active voice in line 133, this suggestion applies more broadly to improve manuscript readability. Multiple sections would benefit from active voice construction, particularly when describing researcher actions. Although passive voice serves appropriate functions (e.g., general statements or when the actor is unknown/insignificant), active voice typically enhances clarity in methodological descriptions. While not essential for approval, consistent voice improvement would enhance reader comprehension.

Response: We appreciate the reviewer’s thoughtful recommendation regarding the broader use of active voice to enhance readability. While we have incorporated some revisions in this direction, we understand that consistent application throughout the manuscript could improve clarity, especially in methodological descriptions.

If the editorial team deems it necessary, we are fully open to implementing these changes during the proof stage, in alignment with PLOS ONE’s final formatting and editorial guidance.

Best regards,

The authors.

---

## [Decision Letter · Decision Letter 2]

10 Aug 2025

Consumption of sorghum-based products and their impact on blood glucose and satiety in adult patients: a protocol for systematic review and meta-analysis

PONE-D-25-15492R2

Dear Dr. Lopes-Júnior,

We’re pleased to inform you that your manuscript has been judged scientifically suitable for publication and will be formally accepted for publication once it meets all outstanding technical requirements.

Kind regards,

Shailender Kumar Verma, Ph.D.

Academic Editor

PLOS ONE

Additional Editor Comments (optional):

Reviewers' comments:

Reviewer's Responses to Questions

**Comments to the Author**

1. Does the manuscript provide a valid rationale for the proposed study, with clearly identified and justified research questions?

Reviewer #3: Yes

2. Is the protocol technically sound and planned in a manner that will lead to a meaningful outcome and allow testing the stated hypotheses?

Reviewer #3: Yes

3. Is the methodology feasible and described in sufficient detail to allow the work to be replicable?

Reviewer #3: Yes

4. Have the authors described where all data underlying the findings will be made available when the study is complete?

Reviewer #3: Yes

5. Is the manuscript presented in an intelligible fashion and written in standard English?

Reviewer #3: Yes

6. Review Comments to the Author

You may also provide optional suggestions and comments to authors that they might find helpful in planning their study.

Reviewer #3: The authors have adequately addressed my critical concerns from the previous review. Most importantly, they corrected the major exclusion criteria error that created a fundamental contradiction in their methodology. The revised exclusion criteria now properly excludes only "children and adolescents, and adults with acute infectious conditions," which is consistent with their study rationale, search strategy, and stated objectives to include adults with chronic conditions like diabetes.

The authors have successfully resolved the methodological inconsistency that was my primary concern, and the manuscript is now scientifically coherent and ready for publication consideration

7. PLOS authors have the option to publish the peer review history of their article (what does this mean? ). If published, this will include your full peer review and any attached files.

**Do you want your identity to be public for this peer review?** For information about this choice, including consent withdrawal, please see our Privacy Policy .

Reviewer #3: **Yes: ** Anthony Onde Morada

---

## [Editor Report · Acceptance letter]

PONE-D-25-15492R2

PLOS ONE

Dear Dr. Lopes-Júnior,

I'm pleased to inform you that your manuscript has been deemed suitable for publication in PLOS ONE. Congratulations! Your manuscript is now being handed over to our production team.

Kind regards,

on behalf of

Dr. Shailender Kumar Verma

Academic Editor

PLOS ONE